# The Effect of Atmospheric Chloride Ions on the Corrosion Fatigue of Metal Wire Clips in Power Grids

**Yifeng Zhang** [1,†], **Wei Chen** [2,†], **Hanbing Yan** [1], **Xuefeng Wang** [1], **Hanping Zhang** [1,\*] and **Shijing Wu** [2,\*]

1   Electric Power Research Institute of State Grid Shanxi Electric Power Company, Taiyuan 030029, China
2   The Institute of Technological Sciences, Wuhan University, Wuhan 430072, China
\*   Correspondence: xhp-123@163.com (H.Z.); wsj@whu.edu.cn (S.W.)
†   These authors contributed equally to this work.

**Abstract:** Corrosion fatigue is an important factor that limits the life of grid materials including wire clips. In order to study the effect of corrosion fatigue and to select suitable grid steels, this paper focuses on the corrosion fatigue properties of Q235 carbon steel, Q235 galvanized steel, and 316L stainless steel in the corrosive environments of air, 2wt% NaCl, 5wt% NaCl, and 8wt% NaCl. Through the fatigue test in the corrosive environment, and the surface morphology scanning and microstructure observation of the fracture, the following conclusions are drawn: the three materials are more susceptible to corrosion fatigue in the $Cl^-$ environment, and the higher the $Cl^-$ concentration, the greater the likelihood of fracture caused by corrosion fatigue for these three materials. By analyzing the surface roughness, dimples, and cracks in the microstructure, it is found that 316L stainless steel is highly sensitive to $Cl^-$ corrosion under cyclic stress, and Q235 galvanized steel is more resistant to $Cl^-$. By plotting the stress fatigue life curve of Q235 galvanized steel, it is found that the corrosion fatigue life decreases as the $Cl^-$ concentration increases. For wire clips in areas with severe $Cl^-$ pollution, Q235 galvanized steel should be selected to achieve the best anti-corrosion fatigue effect; at the same time, the original parts should be repaired or replaced in a timely manner based on the predicted corrosion fatigue life.

**Keywords:** corrosion fatigue; chloride ions; steel fatigue; wire clips; power grids



## 1. Introduction

Metal corrosion causes huge economic losses every year [1,2], and the losses caused by metal corrosion in industrialized countries reach 3% to 4% of the annual gross domestic product (GDP) [3]. With the rapid development of China's economy, the power transmission network has become increasingly large and complex. As the main material comprising the power grid, the amount of metal used has also increased with the development of the power grid system. Power grid accidents caused by severe metal corrosion have become one of the important reasons to implement safe operation of power grids [4,5]. In the power grid system, wire clips with steel as the main material are used as important components to connect transmission lines and electrical equipment [6,7]. As a connection in the power grid, the load carried by overhead transmission lines due to their own gravity, wind, and ice cover [8,9] is applied to the wire clips, meaning that they are subjected to huge alternating stresses.

The wire clips work mainly under atmospheric conditions, and when exposed to the atmospheric environment for a long time, atmospheric corrosion becomes the most important factor causing corrosion [10,11]. Meteorological factors such as temperature, relative humidity, and exposure time affect the specific effects of atmospheric corrosion [12]. Atmospheric pollutants such as $SO_2$ and $Cl^-$ also play an important role in atmospheric corrosion [13]. The combustion of coal is the main source of atmospheric $Cl^-$ on land, and droplets emitted from thermal power plants carry $Cl^-$, which form fine particles of chloride salts in the atmosphere and are deposited on the surfaces of metallic materials under certain

humidity conditions [14–16]. In Shanxi Province, the concentration of $Cl^-$ in particulate matter is 8.31–19.04 $\mu g \cdot m^{-3}$ in winter and 4.28–9.17 $\mu g \cdot m^{-3}$ in summer [17]. The $Cl^-$ penetrates the surface of the metal material into the substrate, causing the metal to undergo electrochemical corrosion and form unstable iron chloride compounds [18], and $Cl^-$ is absorbent and more likely to form an electrolyte of chloride, which is more aggressive toward metals [19]. Pitting occurs in metal under the $Cl^-$ corrosion environment, which can develop into cracks at low cyclic stress (well below the fatigue limit of pure mechanical fatigue), which eventually leads to the metal cracking and failing [20].

Research reports concerning $Cl^-$ on the corrosion of grid metal clips are relatively few, and the research on its corrosion effect is mainly focused on metal materials. Grgur et al. [21] investigated the initial corrosion behavior of AZ63 magnesium alloy exposed to the corrosion media in 1wt%, 3wt%, 5wt%, and 7wt% NaCl solutions, and the results showed that the increase in chloride concentration provokes an increase in the corrosion rate. Kang et al. [22] investigated the effect of $Cl^-$ concentration on stress corrosion cracking (SCC) susceptibility of high manganese steel by immersion tests in different NaCl concentration solutions, and the results showed that the SCC susceptibility of the test steel first increased and then decreased with the increase in $Cl^-$ concentration. Ning et al. [23] studied the crevice corrosion behavior of Alloy 690 in high-temperature aerated chloride solution by SEM, EDS, XRD, and XPS analyses, and the results indicated that the oxide films outside the crevice consisted of Ni–Cr oxides containing a small amount of hydroxides, and the oxide films on the crevice mouth consisted of a $(Ni,Fe)(Fe,Cr)_2O_4$ spinel oxide outer layer and a $Cr(OH)_3$ inner layer, and the oxide films inside the crevice consisted of a $\alpha$-CrOOH outer layer and a $Cr(OH)_3$ inner layer. Shuwei Guo [24] investigated the corrosion behavior and mechanism of Hastelloy C2000 and Inconel 740 Ni-based alloys in chloride-containing supercritical water oxidation, and the results showed that the synergistic effect of chloride and oxygen caused pits on oxides and promoted the formation of $Fe_2O_3$, and the chloride ions had passed through the outer oxide layer and penetrated the inner substrate.

In actual work, we found that the metal clips of the power grid, when used in an environment of industrial air pollution, are often subjected to the combined action of alternating stress and $Cl^-$ corrosion, which leads to the cracking failure of the metal clips, and the service life is shorter than the expected design life. In order to study the service life of metal clips under $Cl^-$ corrosion, this study investigated the corrosion fatigue characteristics of different metal materials (Q235 carbon steel, Q235 galvanized steel, and 316L stainless steel) in different concentrations of $Cl^-$ (2wt% NaCl, 5wt% NaCl, and 8wt% NaCl [25]) through experiments. By fitting the experimental data, the corrosion fatigue life curve of metal materials was obtained and the corrosion fatigue life of metal materials was analyzed, the results of which provide a reference for the design and selection of metal clips.

## 2. Experimental Section

Based on the fact that steel is often used as the main material for wire clamps, this study investigated the actual working environment of a local power grid wire clamp to conduct relevant experiments. The corrosion fatigue performance of different types of steel in $Cl^-$ solutions with different concentrations is mainly investigated, and the fracture morphology and state are obtained to provide a reference for corrosion fatigue life curve and corrosion fatigue life prediction.

### 2.1. Experimental Specimens

In this experiment, Q235, galvanized Q235, and 316L stainless steel bars were selected as specimens. The basic compositions of the three materials are shown in Tables 1 and 2. The shape and size of the specimens were determined according to the requirements in ISO6892-1:2019 [26], and the specific shapes and sizes are shown in Figure 1.

**Table 1.** Chemical composition, yield strength, and tensile strength of Q235.

| Element | C | Mn | Si | S | P | Fe | Yield Strength | Tensile Strength |
|---|---|---|---|---|---|---|---|---|
| Mass/% | 0.18 | 0.5 | 0.2 | 0.03 | 0.04 | 99.05 | 235 MPa | 420 MPa |

Data from the metal material specimen supplier.

**Table 2.** Chemical composition, yield strength, and tensile strength of 316L stainless steel.

| Element | C | Si | Mn | P | S | Cr | Ni | Mo | Fe | Yield Strength | Tensile Strength |
|---|---|---|---|---|---|---|---|---|---|---|---|
| Mass/% | 0.02 | 0.8 | 1.75 | 0.04 | 0.02 | 17.5 | 14.3 | 2.47 | 63.1 | 170 MPa | 480 MPa |

Data from the metal material specimen supplier.

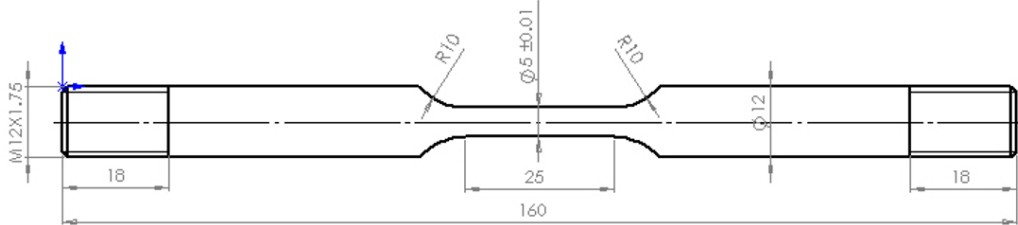

**Figure 1.** Shape and size of the corrosion fatigue specimens.

Based on the atmospheric concentration of $Cl^-$ in the particulate matter described above, the three materials were soaked in 2wt%, 5wt%, and 8wt% NaCl solutions and air (as a control group) for 24 h prior to conducting the experiments, to simulate a corrosive environment.

*2.2. Experimental Instruments*

2.2.1. Corrosion Fatigue Test Equipment

The metal corrosion fatigue performance experiments were carried out using an Instron 8801 compact electrical servo fatigue test system, and the specific parameter settings are shown in Table 3.

**Table 3.** Parameters of the Instron 8801 fatigue testing machine.

| Model | Instron8801 |
|---|---|
| Load range | ±100 KN |
| Effective stroke | 150 mm |
| Accuracy class | 0.5 level |
| Load waveform | Sine, Square, Triangle, Trapezoid |
| Frequency range | 0.01–50 Hz |
| Load relative error | ±0.5% |
| Displacement relative error | ±0.5% |

Data from the experimental instrument parameters.

The test system includes a corrosion fatigue lifting test bench, a high-pressure oil pump, a circulating cooling system, and a computer control system, as shown in Figure 2.

2.2.2. Fracture Morphology Characterization Scanning Equipment

After the specimen was fractured, the fracture morphology was analyzed using a NanoFocus confocal laser microscope from Germany, which uses CCD photography and multi-aperture confocal technology for 3D morphology scanning. The microstructure of the fracture was observed using a MIRA3 high-performance field emission scanning electron microscope. The above two devices are shown in Figure 3.

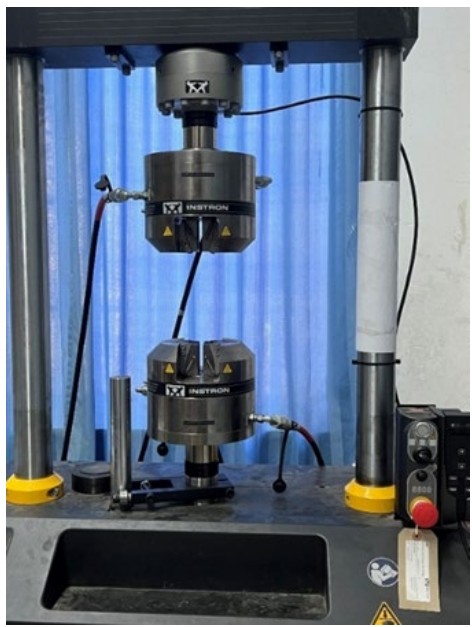

**Figure 2.** Instron 8801 electro-hydraulic servo fatigue testing machine.

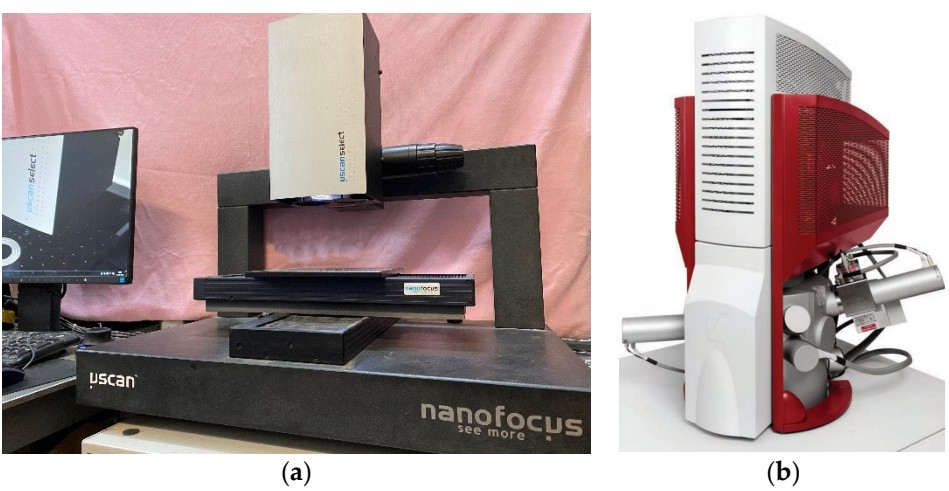

(**a**)                  (**b**)

**Figure 3.** Fracture morphology characterization scanning equipment: (**a**) NanoFocus confocal laser microscope; (**b**) MIRA3 high-performance field emission scanning electron microscope.

### 2.3. Experimental Methods

All experiments were carried out on the Instron 8801 instrument. The parameters of the experimental equipment were set as shown in Table 4, and all experimental procedures were carried out at room temperature. For Q235 carbon steel and 316L stainless steel, using displacement control, the displacement amplitude was set to ±0.02 mm. In order to study the relationship between the fatigue times and stress of Q235 carbon steel, the Q235 galvanized steel specimens were controlled by the load; the stress ratio was set to R = −1; the loading values were set to 7000, 7250, 7500, 7750, and 8000 N; the loading frequency was set to 20 Hz; and the loading waveform was set to sine wave. In order to simulate the actual environment, corrosion continued to occur during the fatigue test, and a suitable Plexiglass container was created. After assembling the specimens within the container, the corrosion solution was added.

**Table 4.** Corrosion fatigue test parameter settings.

| Number | Materials | Control Method | Corrosive Environment | Settings |
|:---:|:---:|:---:|:---:|:---:|
| 1 | | | Air | |
| 2 | Q235 carbon steel | Displacement control | 2wt% NaCl | 0.02 mm |
| 3 | | | 5wt% NaCl | |
| 4 | | | 8wt% NaCl | |
| 5 | | | Air | |
| 6 | Q235 galvanized steel | Load control | 2wt% NaCl | 7000, 7250, 7500, 7750, 8000 N |
| 7 | | | 5wt% NaCl | |
| 8 | | | 8wt% NaCl | |
| 9 | | | Air | |
| 10 | 316L stainless steel | Displacement control | 2wt% NaCl | 0.02 mm |
| 11 | | | 5wt% NaCl | |
| 12 | | | 8wt% NaCl | |

Data from the experimental parameters.

### 2.4. Experimental Process

Before the experiments were conducted, the three steel specimens were polished with 800, 1000, 1200, 1500, and 2000# sandpaper to remove surface impurities, and then washed with deionized water and acetone and air-dried. The three specimens were soaked in 2wt%, 5wt%, and 8wt% NaCl solution and air (as a control group) for 24 h before conducting the experiments. We mounted the specimen assembled in the previous step on the chuck for the experiment. Before clamping the specimens, we set the control program to turn on the specimen protection function to prevent the specimen from being crushed by the chuck.

Afterward, the control software Instron Console of the Instron fatigue test system was used for tuning to measure the stiffness of the specimens with different mechanical properties. During tuning, the waveform was set to a sine wave and the amplitude and frequency were set. The amplitude should not be set too high, to avoid exceeding the elastic deformation limit of the specimens. Additionally, the amplitude should not be too small as this would render the data immeasurable. Therefore, spare specimens should be used for testing.

After completing the tuning, the WaveMatrix2 software was used to establish the fatigue test method. In this experiment, two methods of displacement control and load control were used, both waveforms were sinusoidal, and the loading frequency was set to 20 Hz. For the experiments with displacement control, the displacement amplitude was set to $\pm0.02$ mm. After the preparation, the test was performed and the WaveMatrix2 software interface showed the experimental procedure. The number of fatigue times at fracture was displayed in the software interface, together with the frequency parameters set for the experiment. The change information on the fixture position, load, maximum and minimum displacement, and maximum and minimum load with time can also be observed through this software. After the experiment, the fractures in the specimens were cleaned with absolute ethanol and air-dried to keep the fracture section clean.

### 3. Results

After the corrosion fatigue test, the fracture morphology was scanned with a laser confocal microscope, the microscopic morphology was obtained with an electron microscope, and the specimens were further analyzed. In addition, the stress fatigue life curves were plotted for Q235 galvanized steel taking load control in order to further analyze the effect of different concentrations of NaCl on corrosion fatigue.

### 3.1. Fracture Morphology Analysis

The specimen fractures were analyzed morphologically using a NanoFocus laser confocal microscope from Germany. Firstly, the specimen was fixed on the fixture; secondly, the operating system was used to focus on the specimen to select the appropriate part for scanning; and, finally, the surface topography and surface roughness parameters were selected for comparative analysis.

### 3.1.1. Fracture Morphology Analysis of Q235 Carbon Steel Corrosion Fatigue in Air and 5wt% NaCl

The corrosion fatigue fracture surface morphology of Q235 carbon steel in air and 5wt% NaCl is shown in Figure 4. From the various color distribution areas, the distribution areas in air are more chaotic and interlaced between the different colors, while the boundaries between the various areas in 5wt% NaCl are more gentle and natural, which indicates that during stress loading, the material internally relaxes by plastic deformation to reduce energy and the original grains break up, leading to unevenness and increased roughness of the fracture surface. Therefore, the fracture in air is a ductile fracture, while the fracture in 5wt% NaCl exhibits brittle fracture characteristics [27].

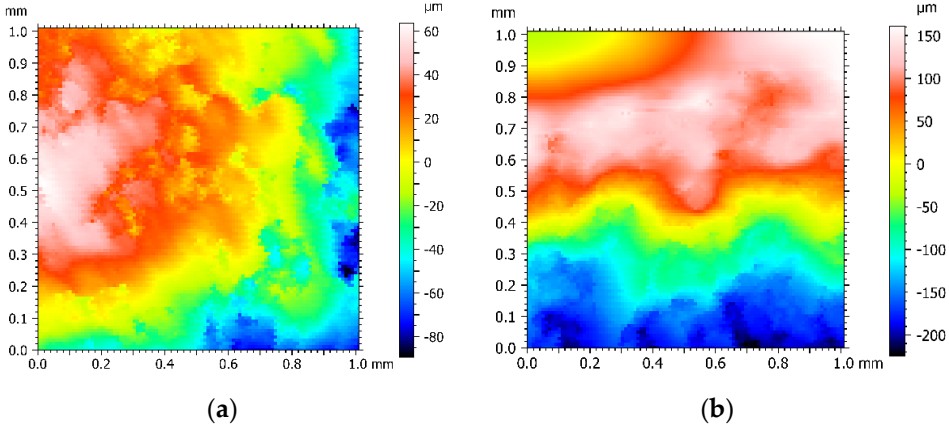

(**a**)                                                                (**b**)

**Figure 4.** Scanning results for Q235 carbon steel: (**a**) surface topography in air; (**b**) surface topography in 5wt% NaCl.

The corrosion fatigue fracture surface roughness of Q235 carbon steel in air and 5wt% NaCl is shown in Table 5. It can be seen from Table 5 that the surface roughness Ra value in air is 2.23 μm, which is significantly larger than the surface roughness of 1.47 μm in 5wt% NaCl. The smooth fracture demonstrates that the corrosion fatigue fracture is more abrupt and the brittle fracture component is higher.

**Table 5.** Fatigue fracture surface roughness of Q235 carbon steel in air and 5wt% NaCl.

| Roughness (Ra)/μm | Q235 Carbon Steel |
|:---:|:---:|
| Air | 2.23 |
| 5wt% NaCl | 1.47 |

Data from the laser confocal microscope.

### 3.1.2. Fracture Morphology Analysis of Q235 Galvanized Steel Corrosion Fatigue in Air and 5wt% NaCl

Figure 5 shows the results of the morphology of Q235 galvanized steel in different dimensions. The galvanized layer of Q235 galvanized steel can be used as a protective layer to resist the corrosion by $Cl^-$ to a certain extent. The morphology of the Q235 galvanized steel fracture in 5wt% NaCl, provided in Figure 5a,b, shows that there are obvious steps and faults in the fracture, which may be due to the fact that $Cl^-$ penetrates through the

galvanized layer and erodes various areas differently, resulting in uneven corrosion of the carbon steel. As $Cl^-$ is able to break the dense $Fe(OH)_2$ protective film generated on the metal [28], the electrolyte penetrates into the substrate, thus accelerating the corrosion, which is manifested by the extremely strong penetration of $Cl^-$. $Cl^-$ breaks the dense $Fe(OH)_2$ film by pitting corrosion. A semicircular hole, usually polygonal, is formed where the film is broken. The diameter of these holes can be up to 1 mm and more. Pitting holes diffuse into the film, and the pitting holes are filled with corrosion products. With the increase in corrosion products, pitting holes gradually diffuse in the film, causing the film to crack and eventually disappear, causing the metal to come into direct contact with $Cl^-$, resulting in corrosion [29,30].

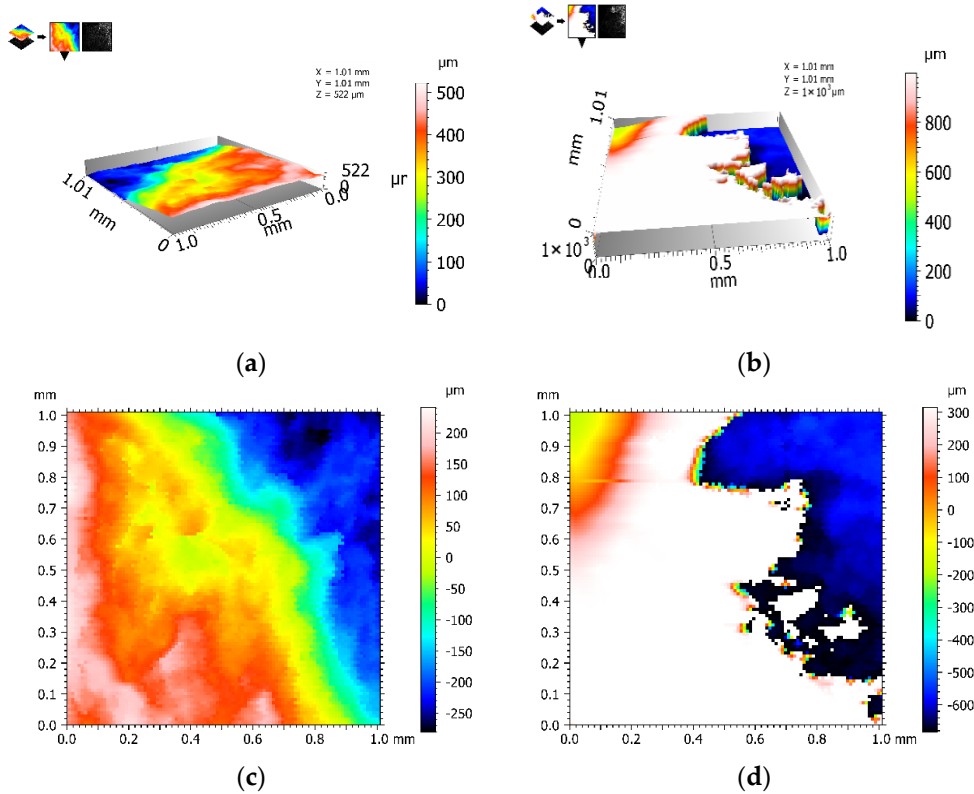

**Figure 5.** Scanning results for Q235 galvanized steel: (**a**) 3D morphology of the air; (**b**) 3D morphology of 5wt% NaCl; (**c**) surface morphology in air; (**d**) surface morphology in 5wt% NaCl.

The corrosion fatigue fracture surface roughness of Q235 galvanized steel in air and 5wt% NaCl is shown in Table 6. It can be seen from Table 6 that the fracture roughness in NaCl is significantly lower than that in air, which also indicates that the fractures in Q235 galvanized steel in 5wt% NaCl exhibit brittle fracture characteristics.

**Table 6.** Fatigue fracture roughness of Q235 galvanized steel in air and 5wt% NaCl.

| Roughness (Ra)/μm | Q235 Galvanized Steel |
|---|---|
| Air | 30.2 |
| 5wt% NaCl | 5.23 |

Data from the laser confocal microscope.

### 3.1.3. Fracture Morphology Analysis of 316L Stainless Steel Corrosion Fatigue in Air and 5wt.% NaCl

The corrosion fatigue fracture surface morphology of 316L stainless steel in air and 5 wt% NaCl is shown in Figure 6. It can be seen from Figure 6a that the boundaries among

various color areas are intertwined with chaotic color changes, and in the air, 316 L stainless steel shows mainly ductile fractures. However, Figure 6b shows that under corrosion by Cl⁻, the color transition is flat and the material exhibits a significant brittle fracture, which is the same conclusion obtained for the two materials described above.

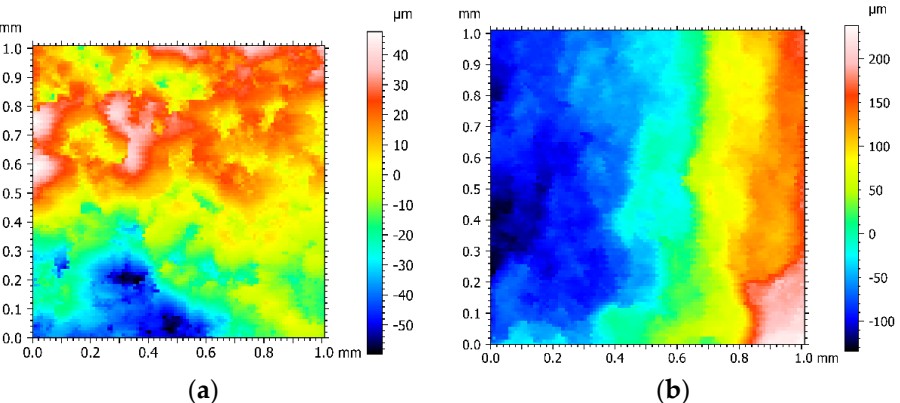

**Figure 6.** Scanning results for 316L stainless steel: (**a**) surface topography in air; (**b**) surface morphology in 5wt% NaCl.

The corrosion fatigue fracture surface roughness of 316L stainless steel in air and 5wt% NaCl is shown in Table 7. The roughness in the air is significantly higher than that in the Cl⁻ environment, which also proves that ductile fractures occurred in the stainless steel in the Cl⁻ environment.

**Table 7.** Fatigue fracture roughness of 316L stainless steel in air and 5wt% NaCl.

| Roughness (Ra)/µm | 316L Stainless Steel |
|:---:|:---:|
| Air | 4.87 |
| 5wt% NaCl | 2.02 |

Data from the laser confocal microscope.

### 3.2. Fracture Microstructure Observation

The scanner can only scan the topography, the display of the fracture is relatively abstract, and the fracture characteristics can only be analyzed by the surface roughness. The images scanned by the electron microscope can reflect the microscopic characteristics of the fracture visually, so that the effect of Cl⁻ corrosion on the material properties can be further analyzed.

#### 3.2.1. Center Morphology Analysis of Q235 Carbon Steel Corrosion Fatigue Fractures in Different Corrosive Environments

The center morphology of Q235 carbon steel corrosion fatigue fractures in different corrosive environments is shown in Figure 7. As can be seen in Figure 7a,b, in air, the cracks in Q235 carbon steel emerge mainly from near the surface and exhibit a certain ductility. It can also be found that many large and deep, tough nests appear on the surface of the material, exhibiting strong plasticity, which is characteristic of a ductile fracture [31]. It can be seen from the fracture image in the Cl⁻ environment that the dimples almost disappear and the characteristics of transgranular cleavage fracture appear, that is, brittle fractures with rapid crack development. As the Cl⁻ concentration increases (see Figure 7b–d), the fatigue fracture mode changes, and the surface crack size gradually increases from small cracks to large ones. The Cl⁻ concentration changes the mechanism of fatigue fracture by forming continuous cracks inside the metal, which causes the specimen to fracture rapidly along the internal cracks.

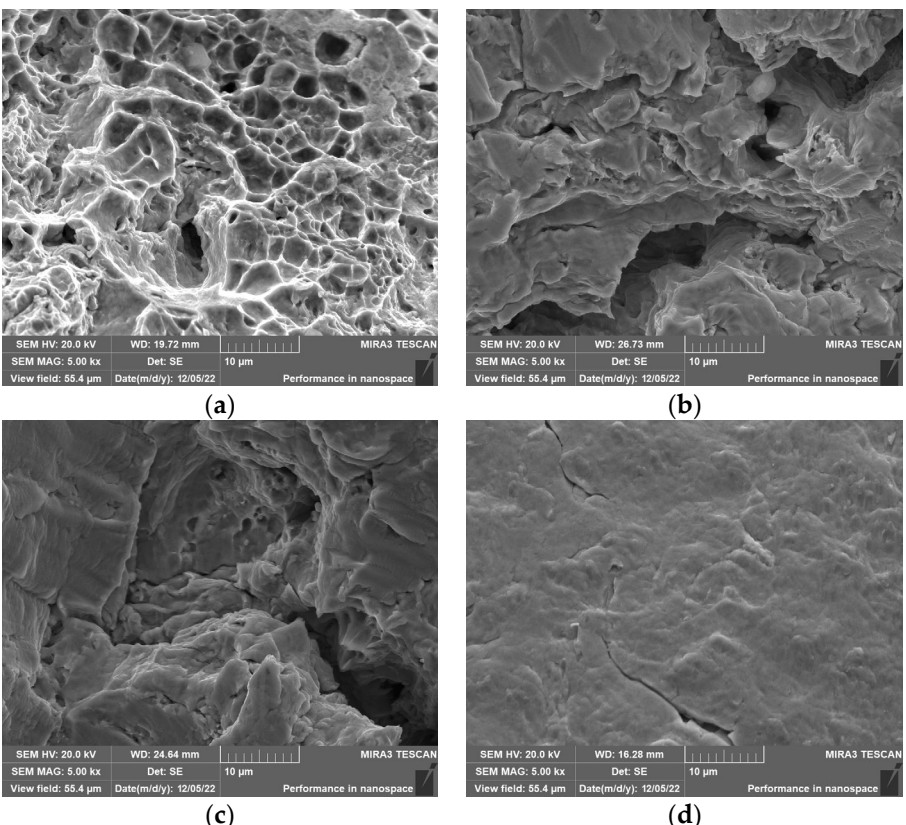

**Figure 7.** Center morphology of Q235 carbon steel fatigue fractures in different environments: (**a**) air; (**b**) 2wt% NaCl solution; (**c**) 5wt% NaCl solution; (**d**) 8wt% NaCl solution.

### 3.2.2. Center Morphology Analysis of 316L Stainless Steel Corrosion Fatigue Fractures in Different Corrosive Environments

The center morphology of 316L stainless steel corrosion fatigue fracture in different corrosive environments is shown in Figure 8. From Figure 8a,b, it can be seen that the fracture in the air environment presents lamellar characteristics, rather than the dimples of Q235 carbon steel, which indicates that 316L stainless steel has poor toughness and is prone to brittle fracture. Moreover, in the fractures in the three subsequent solutions, cracks appear at the fractures and the cracks become deeper and longer with the increase in Cl⁻ concentration. The crack enlargement of Q235 carbon steel is not as significant as that of 316L stainless steel during the increase in Cl⁻ concentration. This means that under cyclic stress, 316L stainless steel is more sensitive to the response of Cl⁻ compared to Q235 carbon steel.

### 3.2.3. Center Morphology Analysis of Q235 Galvanized Steel Corrosion Fatigue Fractures in Different Corrosive Environments

Since the Q235 galvanized steel was subjected to fatigue loading experiments under a fixed load, Figure 9 shows the morphologies of the fatigue fractures in Q235 galvanized steel in different concentrations under a fixed load of 8000 N. In the air and 2wt% NaCl environment, the fractures have dimples, indicating that Q235 galvanized steel still forms ductile fractures in the 2wt% NaCl environment. In the 5wt% NaCl environment, no obvious cracks appeared at the fracture and when the NaCl concentration increased to wt8%, cracks began to appear at the fracture. This indicates that galvanized steel can enhance the Cl⁻ corrosion resistance of metal materials, and Q235 galvanized steel shows better corrosion fatigue characteristics.

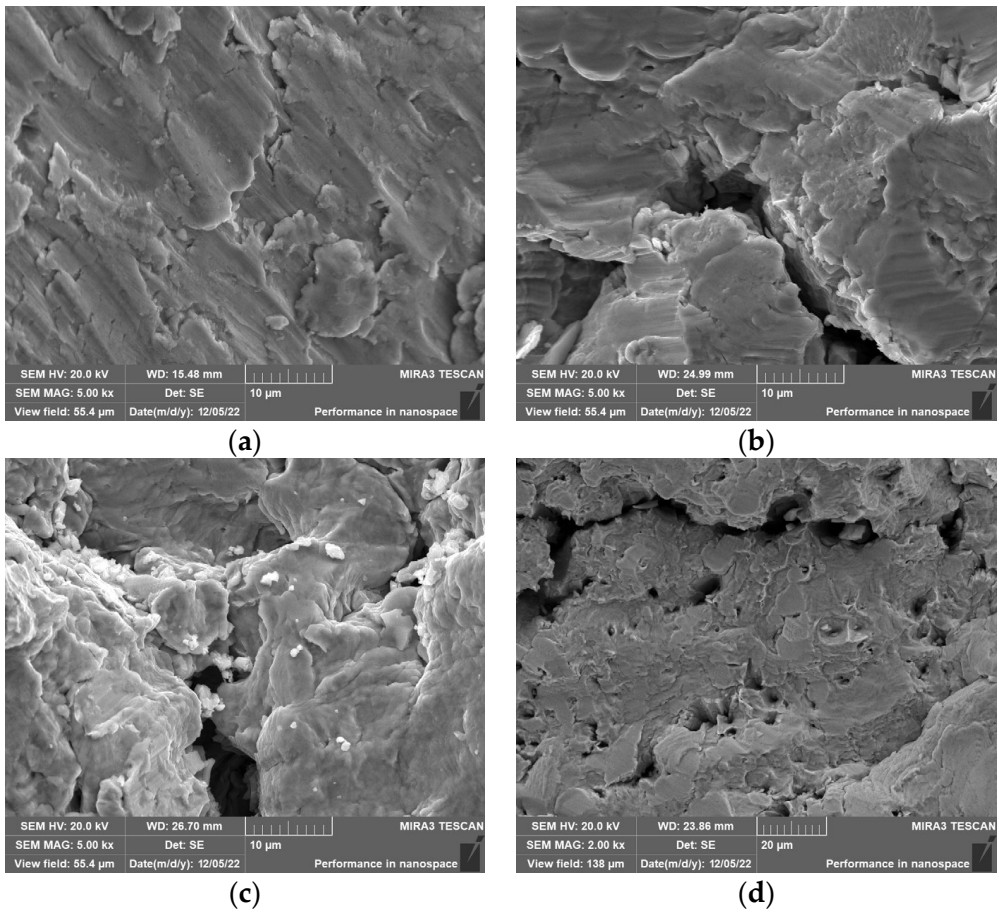

**Figure 8.** Center morphology of 316L stainless steel fatigue fractures in different environments: (**a**) air; (**b**) 2wt% NaCl solution; (**c**) 5wt% NaCl solution; (**d**) 8wt% NaCl solution. The scale in (**d**) is 20 μm.

Through the analysis described above, it is found that the three materials show differences in their susceptibility to $Cl^-$ corrosion. In air, the fracture in 316 L stainless steel shows stronger brittle characteristics than the other materials, and the fatigue performance of stainless steel is poor. The cracks in 316L stainless steel are significantly wider and deeper in the gradually enhanced-$Cl^-$ environment, indicating that the stainless steel is highly sensitive to chloride ions under cyclic stress loading. Additionally, in the corrosive environment, the fractures in Q235 galvanized steel show strong fracture toughness and certain corrosion resistance to $Cl^-$ due to the prevalence of dimples observed in the microstructure.

## 4. Effect of Fatigue Life

As a common material for power grid wire clips, the fatigue life of Q235 galvanized steel deserves attention. The corrosion fatigue test of galvanized steel based on a fixed load can serve to analyze the fatigue life through the stress and the number of bending times before fatigue fracture.

Fatigue life is one of the important factors affecting the service time of wire clips. The prediction of fatigue life has important implications for material selection and processing. Regarding the research content of this paper, the study of fatigue life is also very meaningful for studying corrosion fatigue characteristics. In this paper, a stress–strain-based life prediction method is chosen, using the stress fatigue life curve (S-N curve) as the theoretical basis. Based on the corrosion fatigue data for Q235 galvanized steel, the relationship curves between the maximum loading σ and the number of cycles N for Q235 galvanized steel in four environments were fitted using the least squares method in a double logarithmic coordinate system.

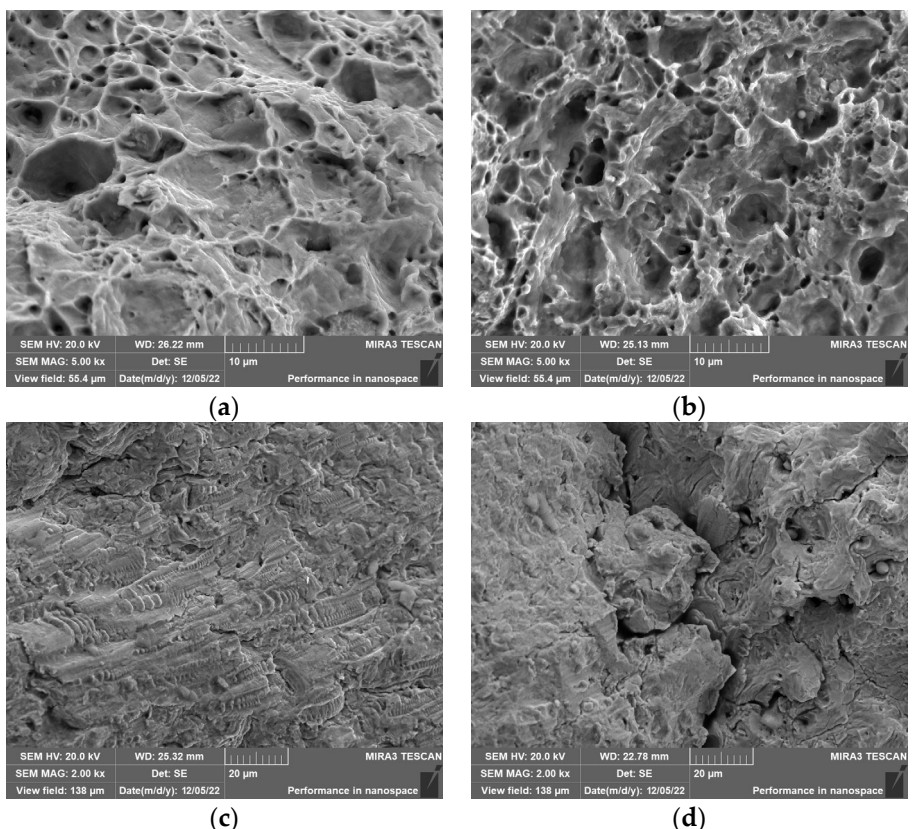

**Figure 9.** Center morphology of galvanized steel fractures under different conditions: (**a**) 8000 N, air; (**b**) 8000 N, 2wt% NaCl solution; (**c**) 8000 N, 5wt% NaCl solution; (**d**) 8000 N, 8wt% NaCl solution. The scale of (**c**,**d**) is 20 μm.

The stress fatigue life curves (S-N curves) of Q235 galvanized steel in the four environments are shown in Figure 10. As the $Cl^-$ concentration increases, the slope of the stress fatigue life curve increases and the curve gradually shifts downward, which means that, under the same load, the number of cycles that the Q235 galvanized steel can withstand decreases and the mechanical properties decrease. Additionally, the fatigue life of Q235 galvanized steel decreases as the load increases in the same corrosive environment.

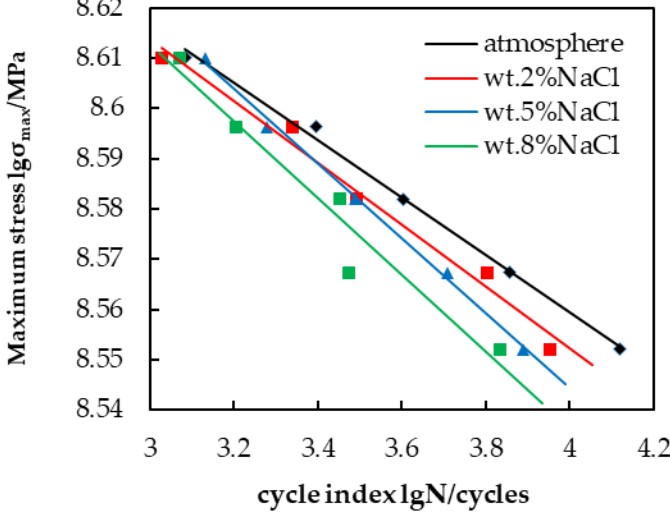

**Figure 10.** Fitting of stress fatigue life curves (S-N curves) for Q235 galvanized steel in four environments.

## 5. Conclusions

The results of corrosion fatigue tests on Q235 carbon steel, Q235 galvanized steel, and 316L stainless steel indicated that dimples at the fracture were reduced in $Cl^-$ environments, and showed obvious brittle fracture characteristics. With the increase in $Cl^-$ concentration, the crack size gradually increased, and the higher the concentration of $Cl^-$, the more severe the corrosion fatigue of the material. The corrosion resistance of 316L stainless steel to $Cl^-$ was the weakest, and the cracks expanded significantly with the increase in $Cl^-$ concentration. However, Q235 carbon steel had strong corrosion resistance to $Cl^-$, and only a few tiny cracks appearing in the fracture. In contrast, Q235 galvanized steel had the strongest corrosion resistance to $Cl^-$, and there were no obvious cracks in 5wt% NaCl solution, which indicated that it could resist the $Cl^-$ corrosion well. Therefore, for the metal clips used in a $Cl^-$ environment, Q235 galvanized steel should be selected as the material to achieve the best anti-corrosion effect and prolong the service life of metal clips.

The stress fatigue life curves (S-N curves) of Q235 galvanized steel showed that the fatigue life of Q235 galvanized steel was shortened with the increase in stress or $Cl^-$ concentration. Therefore, the service life of the metal clips could be predicted by the stress fatigue life curves, and the metal clips can be repaired or replaced in a timely manner according to the predicted corrosion fatigue life.

**Author Contributions:** Conceptualization, Y.Z. and W.C.; methodology, H.Y. and W.C.; validation, X.W. and H.Z.; formal analysis, H.Z. and S.W.; investigation, H.Y.; resources, H.Z.; data curation, W.C.; writing—original draft preparation, W.C.; writing—review and editing, S.W. All authors have read and agreed to the published version of the manuscript.

**Funding:** This research received no external funding.

**Institutional Review Board Statement:** Not applicable.

**Informed Consent Statement:** Not applicable.

**Data Availability Statement:** Not applicable.

**Acknowledgments:** All experimental instruments were obtained from the School of Power and Mechanical Engineering, Wuhan University.

**Conflicts of Interest:** The authors declare no conflict of interest.

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
