# Peer review of "The Effect of Atmospheric Chloride Ions on the Corrosion Fatigue of Metal Wire Clips in Power Grids"

_atmosphere, doi:10.3390/atmos14020237_

Round 1

Reviewer 1 Report

This paper studied the corrosion fatigue performance of Q235 carbon steel, Q235 galvanised steel and 316L stainless steel in four corrosive environments: air, wt.2% NaCl, wt.5% NaCl and wt.8% NaCl. The effect of chloride concentration on corrosion fatigue of three materials was studied by fatigue test in corrosive environment, surface morphology scanning of fracture surface and microstructure observation. By drawing the stress fatigue life curve of Q235 galvanized steel, the variation rule of corrosion fatigue life with Cl concentration is summarized. This work will definitely add valuable knowledge to the current research community, and hence deserves to be published. Nevertheless, the authors should address the following issues before its acceptance for publication.

1.    In the introduction section, a brief overview of the current state of research on chloride ion corrosion of metal wire clamps should be provided, while the introduction needs to be more informative.

2.    The innovative points of this article are not clear enough, please give a more detailed explanation.

3.    In Section 3.1, why only three materials in air and wt.5% NaCl corrosive environments are analyzed? What are the fracture profiles of the three materials in wt.2% NaCl and wt.8% NaCl corrosive environments?

4.    In the references section, the information in references [7] and [13] is incompletely expressed, please complete them.

5.    Line 17, the three materials are 17more susceptible to corrosion fatigue in the Cl- environmentshould be The three materials are 17more susceptible to corrosion fatigue in the Cl- environment.

6.    The conclusion section could be more concise and structured.

7.      The language used in this article needs to be improved in general.

Reviewer 2 Report

In this work, the corrosion fatigue properties of Q235 carbon steel, Q235 galvanized steel and 316L stainless steel in air, wt.2% NaCl, wt.5% NaCl and wt.8% NaCl corrosion environment were studied. It is found that the corrosion fatigue life decreases with the increase of Cl- concentration. This indicates that Q235 galvanized steel should be selected for wire clips in Cl- polluted areas to achieve the best anti-corrosion fatigue performance. The research content of this work is innovative and complete. But there are still some problems. I have several concerns and questions that need proper attention and should be well addressed:

1.Line 18: The author can unify all the Cl- writing formats in the text

Results and discussions

2.Table 5: The scanning area of laser confocal microscope is not representative, and the roughness value needs to be determined by the weighted average of a large number of data.

3.Line 182: “As Cl- is able to break the dense Fe(OH)2 protective film generated on the metal” Literature is needed here.

4.Line 189: The fracture mechanism can not only be confirmed by the port roughness value, but also need other auxiliary means, such as SEM, etc.

5.Line 245-250: What is the effect law of different concentration of NaCl on the corrosion fatigue performance of Q235 galvanized steel?

6.The very latest references about the properties of alloys (e.g., ACS Appl. Mater. Interfaces 2021, 13, 55712−55725; Acta Materialia 232 (2022) 117934; Friction 10(11): 1913–1926 (2022)) are lacking, which may be helpful for introduction and/or discussion.

7.Electrochemical corrosion in Cl- environment is also one of the important reasons for material failure. The author can further investigate the electrochemical corrosion performance of the three materials to further prove that Q235 galvanized steel was fit for using as the wire clips material in the Cl- polluted area.

Reviewer 3 Report

In the manuscript entitled “The Effect of Atmospheric Chloride Ions on the Corrosion Fatigue of Metal Wire Clips in Power Grids” by Yifeng Zhang, Wei Chen, Hanbing Yan, Xuefeng Wang, Hanping Zhang, and Shijing Wu the authors compare the fatigue response of different alloys in a full immersion chloride environment. Through various techniques including confocal microscopy and SEM, the authors conclude that the galvanized Q235 alloy is the most resistant to corrosion fatigue. While various trends were observed across alloys,  the data is incomplete, is not put within the context of literature, and there are no replicates presented.. Based on this, rejection of the article is suggested. Comments are outlined below with critical concerns marked by “**,” major concerns being marked by “*,” and minor concerns left unmarked.

Introduction

·         *Page 2 Line 51-54: what is the corrosion layer? Is this an oxide film or corrosion product? Please specify. Additionally, reference 18 is not about atmospheric corrosion rather, is about molten salt chlorides. Are the mechanisms the same?

·         Page 2 line 56: the minus on chloride needs a superscript

Experimental

·         **Table 1 and 2: These compositions are insufficient. The ranges are most likely pulled from a standard and do not include Fe. Actual compositions of the materials should be noted either from additional analysis or from certificates of composition from the supplier.

·         Figure 3: showing images of the analysis equipment does not compliment the data nor the overall story

·         **Table 4: How can a comparison be made between the carbon steel/SS316 and the galvanized steel when the same tests were not ran on all material types.

·         **Page 4 Lines 111-114: Full immersion most likely will not represent “real environment, corrosion”. The tests were performed under full immersion, and are not representative of an atmospheric environment where the solution composition and concentration is a function of the RH. Additionally, thin brine layers (< 800 um) are likely in atmospheric environments and are a strong function of salt load.

·         *Page 5 Lines 120-141: The grammar in this portion is completely different than the rest of the paper and distracts from the flow of the paper

Results

·         **Section 3.1.1: It is suggested that optical images should be shown with the surface topography. Additionally, the scales for the images should be normalized or it should noted that the scales are different. Are the scans of the whole fracture surface, final failure, or just a selective section of the material? The area selection could highly skew the results. As such, gaining mechanistic interpretations from the profilometry scans would be highly challenging and there is insufficient evidence at this point to draw conclusions

·         **Section 3.1.2: There is a lot of interpretation in this section with no references and therefore is not a result. There is no result that displays an influence of the galvanized steel, Fe(OH)2, or an influence of Cl penetration. Further, in figure 5(b) the profilometry scans are incomplete and run into the limit of the machine. It is not possible to ascertain the behavior of the fracture surface due to this. Additionally, the surface roughness results inaccurate due to the scanning of the metal surface.

·         *Page 8 Lines 22-226: the discussion of fracture mode is not consistent with existing literature. While there is a clear change in fracture morphology from (b) to (d) the description is insufficient to describe the behavior. Further, where were these images taken from? Was it a similar spot on all samples?

·         *Page 9 Lines 235-240: Similar comment as the carbon steel material but how can one tell if the cracks are “deeper and longer”. The “cracks” should be highlighted in the micrograph. Can the fatigue striations be seen in the micrographs? Do the striations change with chloride concentrations?

·         **Page 10 Lines 254-261: As previously stated, the tests were ran under different conditions and therefore it is inappropriate to compare between alloys.

Conclusions

·         **Page 11 Lines 286-288: Per the profilometry data, the surface roughness was reduced in comparison to the air experiments?

·         **Overall Conclusions: Per the previous comments, many of the data presented could be skewed, incomplete, or different tests were ran between samples. This is a major concern in terms of the data shown. Further, replicates are not ran nor are various portions of the fracture surface explored especially at different magnification levels to investigate potential mechanisms present.

Reviewer 4 Report

The manuscript studied on the Effect of Atmospheric Chloride Ions on The Corrosion Fatigue of Metal Wire Clips in Power Grids . The results and discussion need be further to a detailed analysis. Meanwhile, there are some mistakes throughout this manuscript. I recommend Reconsider after major revision. The main comments are listed as below. 

1.The introduction does not present the scientific problems. This paper need make a theoretical analysis of some recent literature in the past and clarify the highlights of their own research.

2.The mechanism of the Effect of Atmospheric Chloride Ions on The Corrosion Fatigue of Metal Wire Clips need to be further discussed.

3. It need to provide the Corrosion fatigue fitting curve.

4. The author need to explain why used the 2wt.%, 5wt.%, 8wt.% NaCl solutions to simulate a corrosive environment.

Round 2

Reviewer 3 Report

In the manuscript entitled “The Effect of Atmospheric Chloride Ions on the Corrosion Fatigue of Metal Wire Clips in Power Grids” by Yifeng Zhang, Wei Chen, Hanbing Yan, Xuefeng Wang, Hanping Zhang, and Shijing Wu the authors compare the fatigue response of different alloys in a full immersion chloride environment. Through various techniques including confocal microscopy and SEM, the authors conclude that the galvanized Q235 alloy is the most resistant to corrosion fatigue. While various trends were observed across alloys,  the data is incomplete, is not put within the context of literature, and there are no replicates presented.

Further, many of the concerns in the first review were not addressed and have been left below. There are still errors in the profilometry (Figure 5(b)) that can skew data analysis, and conclusions. It can be seen directly from the image that the scans were not taken appropriately and are not quality for publication. It is still unclear how the profilometry images were taken, what part of the fracture surface they represent, and distract from the overall message. Finally, the discussion of Cl- effects on corrosion and corrosion fatigue are incomplete and inconsistent with literature.

Based on this, rejection of the article is suggested. Comments are outlined below with critical concerns marked by “**,” major concerns being marked by “*,” and minor concerns left unmarked.

Introduction

Experimental

·         Figure 3: showing images of the analysis equipment does not compliment the data nor the overall story

·         **Table 4: How can a comparison be made between the carbon steel/SS316 and the galvanized steel when the same tests were not ran on all material types.

·         **Page 4 Lines 136: Full immersion most likely will not represent “real environment, corrosion”. The tests were performed under full immersion, and are not representative of an atmospheric environment where the solution composition and concentration is a function of the RH. Additionally, thin brine layers (< 800 um) are likely in atmospheric environments and are a strong function of salt load.

·         *Page 6: The grammar in this portion is completely different than the rest of the paper and distracts from the flow of the paper

Results

·         **Section 3.1.1: It is suggested that optical images should be shown with the surface topography. Additionally, the scales for the images should be normalized or it should noted that the scales are different. Are the scans of the whole fracture surface, final failure, or just a selective section of the material? The area selection could highly skew the results. As such, gaining mechanistic interpretations from the profilometry scans would be highly challenging and there is insufficient evidence at this point to draw conclusions

·         **Section 3.1.2: There is a lot of interpretation in this section with no references and therefore is not a result. There is no result that displays an influence of the galvanized steel, Fe(OH)2, or an influence of Cl penetration. Further, in figure 5(b) the profilometry scans are incomplete and run into the limit of the machine. It is not possible to ascertain the behavior of the fracture surface due to this. Additionally, the surface roughness results inaccurate due to the scanning of the metal surface.

o   How does chloride “break the dense Fe(OH)2 film”? Does it diffuse through the film? Does it cause the film to rupture? There is a lot of research on this area (not just for galvanized steel, that could be used to help with the mechanistic interpretations.

·         *Page 8: the discussion of fracture mode is not consistent with existing literature. While there is a clear change in fracture morphology from (b) to (d) the description is insufficient to describe the behavior. Further, where were these images taken from? Was it a similar spot on all samples?

·         *Page 9-10: Similar comment as the carbon steel material but how can one tell if the cracks are “deeper and longer”. The “cracks” should be highlighted in the micrograph. Can the fatigue striations be seen in the micrographs? Do the striations change with chloride concentrations?

·         **Page 10-11: As previously stated, the tests were ran under different conditions and therefore it is inappropriate to compare between alloys.

Conclusions

·         **Page 12: Per the profilometry data, the surface roughness was reduced in comparison to the air experiments?

·         **Overall Conclusions: Per the previous comments, many of the data presented could be skewed, incomplete, or different tests were ran between samples. This is a major concern in terms of the data shown. Further, replicates are not ran nor are various portions of the fracture surface explored especially at different magnification levels to investigate potential mechanisms present.

·         **Overall: There are also major differences in terms of the material properties (i.e., yield strength, UTS, etc.) that are not account for in this work. How do these play a role in the fracture morphologies and the mechanisms for corrosion fatigue?

·         **Overall: Per comment in the experimental section, the tests were performed in full immersion. Is this representative of atmospheric conditions that the metal clips will be exposed to?

Reviewer 4 Report

Accept.
